# Cerebrospinal Fluid Protein Concentrations in Hydrocephalus

**DOI:** 10.3390/children10040644

**Published:** 2023-03-30

**Authors:** Florian Wilhelmy, Matthias Krause, Stefan Schob, Andreas Merkenschlager, Robin Wachowiak, Wolfgang Härtig, Jürgen Meixensberger, Janina Gburek-Augustat, Tim Wende

**Affiliations:** 1Department of Neurosurgery, University Hospital Leipzig, Liebigstrasse 20, 04103 Leipzig, Germany; 2Division of Neuroradiology, University Hospital Leipzig, Liebigstrasse 20, 04103 Leipzig, Germany; 3Department of Childrens and Adolescence Health, Division of Neuropediatrics, University Hospital Leipzig, Liebigstrasse 20, 04103 Leipzig, Germany; 4Department of Pediatric Surgery, University Hospital Leipzig, Liebigstrasse 20, 04103 Leipzig, Germany; 5Paul-Flechsig-Insitute for Brain Research, Liebigstraße 19, 04103 Leipzig, Germany

**Keywords:** cerebrospinal fluid, CSF protein, hydrocephalus, intracranial pressure, aqueductal stenosis, normal pressure hydrocephalus

## Abstract

CSF protein levels are altered in neurological disorders, such as hydrocephalus of different etiologies. In this retrospective observational study, we analyzed cerebrospinal fluid (CSF) samples in hydrocephalic diseases such as aqueductal stenosis (AQS, *n* = 27), normal pressure hydrocephalus (NPH, *n* = 24), hydrocephalus communicans (commHC, *n* = 25) and idiopathic intracranial hypertension (IIH)/pseudotumor cerebri (PC, *n* = 7) in comparison with neurological patients without hydrocephalic configuration (control, *n* = 95). CSF was obtained through CSF diversion procedures and lumbar punction and analyzed for protein concentrations according to the institution’s laboratory standards. We found significantly decreased CSF protein levels in patients suffering from AQS (0.13 mg/dL [0.1–0.16 mg/dL] *p* = 2.28 × 10^−8^) and from PC (0.18 mg/dL [0.12–0.24 mg/dL] *p* = 0.01) compared with controls (0.34 mg/dL [0.33–0.35 mg/dL]). Protein levels were not altered in patients suffering from commHC and NPH compared with neurologically healthy individuals. We propose that a decrease in CSF protein levels is part of an active counterregulatory mechanism to lower CSF volume and, subsequently, intracranial pressure in specific diseases. Research regarding said mechanism and more specific proteomic research on a cellular level must still be performed to prove this hypothesis. Differences in protein levels between different diseases point to different etiologies and mechanisms in different hydrocephalic pathologies.

## 1. Introduction

Cerebrospinal fluid (CSF) contains a significant number of proteins and other molecules that are crucial for brain function. There is active protein secretion from the choroid plexus to maintain vital functionalities, e.g., transthyretin, as well as significant clearance of molecules to maintain an optimal environment dedicated to guaranteeing neuronal function [1,2]. In-depth proteomics detected more than 2000 proteins in human CSF that have not been found in human plasma [3,4,5]. Albumin contributes to more than 50%, but no more than 75%, of the total CSF protein content. Furthermore, surfactant-associated proteins A–D have recently been characterized and quantified as CNS–inherent proteins produced by the choroid plexus, the ependymal layers and the perivascular spaces as components of human CSF under physiological conditions [6].

An increase in the total protein concentration is usually observed in states of blood–brain barrier (BBB) breakdown during inflammatory or autoimmune reactions but has largely been attributed to changes in CSF flow rate [7]. However, little is known about CSF levels of protein, glucose or lactate in hydrocephalus, where CSF flow would most likely be altered.

Therefore, the aim of this study was to examine the differences in the total protein concentrations in hydrocephalus patients compared with control subjects without hydrocephalic configuration.

## 2. Methods

Patients who underwent CSF diversion procedures were routinely examined for their total CSF protein content at time of surgery. Furthermore, radiological MRI data and clinical data were available, including data from follow-up examinations from at least 3 months afterward. We categorized the hydrocephalus recorded in the literature according to the following pathologies: aqueductal stenosis (AQS), normal pressure hydrocephalus (NPH), communicating hydrocephalus (comm HC) and pseudotumor cerebri (PC). Diagnoses were confirmed radiologically and by clinical presentation. Furthermore, commHC was suggested in patients with signs of chronically increased intracranial pressure without radiological CSF flow obstruction. PC was defined after proof of disease was confirmed by intracranial pressure monitoring and consistent MRI without ventricular enlargement.

CSF specimens from 95 patients without hydrocephalic configuration served as controls. These were obtained during routine neurological diagnostics undertaken with lumbar punction in both neurological and neurosurgical departments and remained without proof of neurological disorders. Lumbar punction was undertaken in these cases to rule out neurological/CSF pathologies. In all patients and controls, CSF infection, preceding trauma or hemorrhage were ruled out. An overview of demographic data of each group is given in Table 1.

Statistical analysis for the groups was performed using Welch’s *t*-test with IBM SPSS Statistics 29 (IBM, Armonk, NY, USA). Significance level was set at 0.05.

All patients or caregivers gave their informed consent for the collection of clinical, CSF and radiological data. A local ethics committee approved the study (Ethikkommission Universität Leipzig Az 330-13-18112013) in November 2013.

## 3. Results

A total of 178 patients were enrolled in this study. In total, 27 patients suffered from AQS, 25 from NPH, 24 from commHC and 7 from PC. The total CSF protein, glucose and lactate concentrations are shown in Table 1. Glucose and lactate concentrations did not reveal any significant differences between the groups. CSF protein concentrations in patients with AQS were substantially decreased in comparison with all other hydrocephalic groups and compared with the control group (*p* < 0.01). Pseudotumor cerebri patients showed a trend of lowered CSF protein concentrations (*p* = 0.01). There were no differences between control and hydrocephalic entities in general (Figure 1).

## 4. Discussion

This retrospective study analyzed perioperative CSF protein concentrations in correlation to the diagnoses of hydrocephalus and compared with control subjects.

It is widely understood and agreed on that the major portion (70–80%) of CSF is formed and secreted in the choroid plexus [8] and the lesser portion (10–20%) crosses the blood–brain barrier [9]. It is in these compartments that the composition of CSF is regulated by epithelial cells [10]. Its resorption and circulation from this point onward, however, is widely debated. The pathways originally described include the flow via the subarachnoid space, the arachnoid villi and drainage into the blood of the sagittal sinus [11]. Recently, other pathways through lymphatic drainage, deep intracranial veins, and spinal lymph nodes have been suggested [12]. It has been observed that CSF protein is distributed in the brain via convective flow and diffusion [13], and the interaction between interstitial fluids and CSF has been described [14], suggesting not only a link between the compartments but also the presence of regulatory mechanisms. Although the main regulatory parameter for CSF secretion is thought to be intracranial pressure, it has been proven that CSF composition can be regulated independently [15]. Osmolarity has been found to influence CSF secretion [16]. The role of CSF ion composition and the in-detail clarification of drainage pathways may be of some importance to proteomics but will be discussed elsewhere [17].

### 4.1. CSF Ciruculation in Underlying Hydrocephalic Pathologies

Although they share some features, the different hydrocephalic entities in the present cohort vary in their pathophysiologies: AQS causes a CSF flow disruption with consecutively raised intracranial pressure, which is more likely to be detected at a young age [18]. PC exhibits no such disruption, but both entities display elevated pressure and narrower ventricles [19]. While pressure is not elevated and the ventricles are wider in NPH, it shows no flow disruption [20]. Communicating HC, by definition, is not accompanied by a flow disruption; however, CSF malabsorption is prevalent in these cases [21]. In summary, the underlying pathologies directly affect CSF circulation and suggest the presence of counterregulatory mechanisms.

### 4.2. Interpretation of Altered Protein Concentrations in AQS and PC Patients

Given that the traditional assumption is that CSF is merely a blood ultrafiltration and that the main regulatory parameter for CSF secretion should be intracranial or intraventricular pressure, differences in CSF proteomics would seem unlikely. However, the reduction in overall CSF protein concentrations in AQS and PC patients compared with controls possibly reflects a counterregulatory mechanism that attempts to reduce intracranial CSF volume and, subsequently, intracranial pressure. Given these assumptions, our findings support the recent conclusions of Krishnamurthy et al. [22], who described hydrocephalus to be influenced by the effective clearance of macromolecules out of the CSF space via the paravascular and lymphatic pathways [22]. The lowering of CSF protein levels in AQS as well as PC patients rules out a dilution effect of enlarged ventricular volume but makes an association with pathological intracranial pressure and/or pulsatility very likely. Since CSF flow differs in these two pathologies, this finding would most likely point toward changes in CSF production. Furthermore, some CSF protein contents, e.g., surface-active agents such as surfactant proteins A and C, have been demonstrated to be significantly increased in states of ventriculomegaly. Whether the changes in protein levels merely reflect a symptom of the underlying pathology has yet to be discarded. Here, our findings remain strictly descriptive. It has to be noted that the changes in osmolarity through the oncotic pressure of the relatively small protein concentrations [23] are not to be expected, which makes protein-driven changes in osmolarity as a regulatory mechanism in pressure pathologies less likely.

### 4.3. Interpretation of Altered Protein Concentrations in NPH and Communicating HC Patients

The reasons explaining the unchanged protein concentrations in NPH and communicating HC patients are unclear, but other abnormalities, and even possible treatment, have been described [24,25]. Impaired counterregulatory reduction might even be a pathophysiological momentum in the genesis of those chronic hydrocephalic states [26]. Moreover, an isolated pathology in CSF resorption, independent of CSF production or flow, would appear possible. Again, this would imply that CSF composition in hydrocephalus is mostly determined in CSF production [27].

### 4.4. Outlook

Further studies with detailed proteomic examinations of CSF components are necessary to further elucidate the nature and mechanisms of the reduction in CSF protein concentrations. These studies will also have to enroll even larger numbers of patients. Additionally, longitudinal sampling during prolonged treatment periods under CSF diversion procedures should be undertaken. A clear distinction between chronic and acute hydrocephalus, clarification of the side of the obstruction, if present, and inclusion of hypersecretory pathologies would further elucidate part of the problem—assuming that CSF protein concentrations are but a symptom. Alternatively, these desiderata would further elucidate the body’s solution to pathological intracranial pressure. We therefore see great diagnostic and therapeutic potential in CSF proteomics.

## 5. Conclusions

The total CSF protein concentrations are significantly lower in patients with aqueductal stenosis and pseudotumor cerebri compared with other chronic hydrocephalic conditions. The nature of this reduction remains unclear and warrants further investigation.

## Figures and Tables

**Figure 1 children-10-00644-f001:**
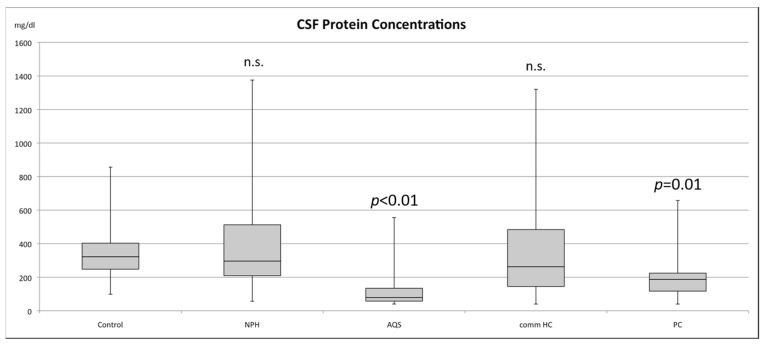
Boxplot for the total CSF protein concentrations for the individual subgroups of hydrocephalic patients, pseudotumor cerebri patients and the control group. *p*-values compared with controls are shown. *p*-values above 0.05 were considered non-significant (n.s.).

**Table 1 children-10-00644-t001:** Overview of demographic data and CSF analysis data for all patients.

	Control	AQS	Communicating HC	NPH	Pseudotumor Cerebri
*n*	95	27	25	24	7
Age (yrs)	42.7	17.3	28.3	69.7	25.7
Sex (m/f)	50/45	18/9	13/12	19/8	1/5
Lac	1.63 (1.62–1.64)	1.50 (1.49–1.51)	1.95 (1.77–2.11)	1.85 (1.77–1.94)	1.53 (1.32–1.74)
Glu	3.8 (3.62–3.98)	3.48 (3.06–3.8)	3.22 (3.01–3.43)	4.10 (3.97–4.33)	3.53 (3.08–3.97)
Tot. Protein	0.34 (0.33–0.35)	0.13 (0.10–0.16)	0.35 (0.27–0.43)	0.37 (0.30–0.44)	0.18 (0.12–0.24)

## Data Availability

All data is primarily available in the manuscript.

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
