# Peer review of "Cerebrospinal Fluid Protein Concentrations in Hydrocephalus"

_children, 2023, doi:10.3390/children10040644_

Round 1

Reviewer 1 Report

Dear Authors

Thank for submiting this study. 

Your study was narrative without a deep discussion especially in physiopathology. You need to give more information about the group controls: you choose it from neurological department? why the need to have a LP? Also more informations about the physiopathology of the result and deep discussion with comparaison about other neurological disorders can help the readers

Author Response

Your study was narrative without a deep discussion especially in physiopathology.

This is entirely true. As a brief report, we focused on our findings. Nonetheless, we overhauled the discussion to give more context.

You need to give more information about the group controls: you choose it from neurological department? why the need to have a LP?

Patients underwent lumbar punction for a great variety of reasons; we chose patients that underwent lumbar punction in "rule-out" diagnostics, that were later cleared from neurological pathologies. These were recruited both from the neurological as well as the neurosurgical department. We specified in the "Methods" section accordingly. 

Also more informations about the physiopathology of the result and deep discussion with comparaison about other neurological disorders can help the readers. 

We added references to the known literature and also extended the discussion. Nonetheless, we struggled to find physiopathological explanation beyond theorizing in the literature, which was key motivation to publish our findings "as is".

Reviewer 2 Report

The manuscript is good, however, the title is not that attractive. The discussion should have more content of explanation for more understanding.

Author Response

The manuscript is good, however, the title is not that attractive.

We are glad you liked the reading. Lacking a better one, we stuck with the title though.

The discussion should have more content of explanation for more understanding.

This was a key point among the reviewers. Although we published the study as a "brief report", we extended the discussion according to your suggestion to provide more context.

Reviewer 3 Report

Thank you for inviting me to review this paper. The authors did a great job studying the protein concentration in hydrocephalus patients. Overall, it is very well prepared and straight to the point. However, it holds some small areas for improvement. Please consider these minor comments:

- The abstract can not stand alone. Methods used were not mentioned at all. It is very uninformative. Study type, data selection, sample size, … must all be stated.

- Authors affiliation is not clear. Please rewrite each, stating the department, institution, city and country

- Which program was used in statistics?

- I suggest that you add the p values in the figure. Over each disease section, put the p value that is found when compared to the control section, and explain that in the figure legend.

- Controls could have been better if specific for each disease due to the big difference in age. For example, average age of controls is 42, that of AQS category was 17 and that of NPH category was 69. Each needs a control group with similar age. However, it is okay if this this comment could not be addressed.

This article offers an addition to the literature. Results are of great importance and add to the field of study. The conclusion well adheres the manuscripts. Results of this study offer a good start for further investigation.

Good Luck

Author Response

Thank you for inviting me to review this paper. The authors did a great job studying the protein concentration in hydrocephalus patients. Overall, it is very well prepared and straight to the point. However, it holds some small areas for improvement. Please consider these minor comments:

  • The abstract can not stand alone. Methods used were not mentioned at all. It is very uninformative. Study type, data selection, sample size, … must all be stated.

The methods have been briefly included. As the abstract is limited to 200 words, we stated the most important facts as suggested: Sample size per group, data selection, and study type.

  • Authors affiliation is not clear. Please rewrite each, stating the department, institution, city and country

So we did.

  • Which program was used in statistics?

IBM, SPSS, was added.

  • I suggest that you add the p values in the figure. Over each disease section, put the p value that is found when compared to the control section, and explain that in the figure legend.

We added p-values or stated "non-significant" in the graphic and explained in the legend accordingly.

  • Controls could have been better if specific for each disease due to the big difference in age. For example, average age of controls is 42, that of AQS category was 17 and that of NPH category was 69. Each needs a control group with similar age. However, it is okay if this this comment could not be addressed.

This a  ery good point indeed. Due to the sample size and the different age groups of certain diseases (e.g. there is no pediatric NPH and AQS is mostly diagnosed in a young age) we were not able to rule out demographic influence completely. We will be trying to do so in the future with matched pairs.

This article offers an addition to the literature. Results are of great importance and add to the field of study. The conclusion well adheres the manuscripts. Results of this study offer a good start for further investigation.

Round 2

Reviewer 1 Report

Dear Authors

thank you for this new version